# A new validated Lymphoedema-specific Patient Reported Outcome Measure (LYMPROM) for adults with Lymphoedema

**Melanie Thomas**[1], **Marie Gabe-Walters**[1], **Ioan Humphreys**[2*], **Alan Watkins**[2]

**1** Lymphoedema Wales Clinical Network, Swansea Bay University Health Board, Wales, United Kingdom,
**2** Faculty of Medicine, Health and Life Sciences, Swansea University, Wales, United Kingdom

\* I.Humphreys@Swansea.ac.uk

## Abstract

### Background

A new lymphoedema-specific Patient Reported Outcome Measure (LYMPROM©) was developed to help patients easily report the impact of their lymphoedema and enable lymphoedema therapists to understand what matters most to patients.

### Aims

This paper describes the validation of LYMPROM© for adults with lymphoedema.

### Methods

A multi-phased iterative review was undertaken to investigate the reliability and validity of LYMPROM©. Face and content validity were reviewed by surveying patient representatives and healthcare professionals, along with the validity of LYMPROM© Cymraeg, a Welsh translation. Following COSMIN guidelines, validation study phases used anonymised routinely collected data to examine internal consistency, structural validity, construct validity (compared with the EQ5D-5L), measurement error, test-retest reliability and responsiveness.

### Results

LYMPROM© demonstrated validity (content and construct) and reliability (test-retest, internal consistency). All items were regarded as relevant, comprehensive and clear, with item content validity index (CVI) between 0.83 to 1.00, and average overall assessment of 0.94. Robust development of LYMPROM© Cymraeg ensured appropriate translation into Welsh. LYMPROM© item scores, with means and medians generally in the lower half of the scale, were positively correlated, as were three (Physical health, Social health and Emotional health) domain scores (domain correlations: 0.595 to 0.812). LYMPROM© total and domain scores showed moderate negative correlations (-0.577 to -0.435) with EQ5D-5L measures. LYMPROM© total and domain scores showed good test-retest (within two weeks) properties, with little or no change in mean or median scores, and strong positive correlations

**Data availability statement:** All data files are available via the Swansea University Open research data Community, hosted by Zenodo: https://zenodo.org/records/14194356

**Funding:** The author(s) received no specific funding for this work.

between test and retest scores (Total: 0.919; Physical health domain: 0.922; Social health domain: 0.889; Emotional health domain: 0.820).

LYMPROM© showed good responsiveness, with strong, positive correlations between total and domain initial and repeat (between four weeks and seven months later) scores, with a slight reduction in scores (-3.8 to -2.0 units) and some indication of relationships between reduction and time interval (Total: p = 0.025; Physical health domain: 0.034; Social health domain: 0.181; Emotional health domain: 0.009).

## Conclusion

Evidence shows that LYMPROM© offers a reliable and valid tool for use in clinical practice. Scores on three domains allow a more granular assessment of the patient's view of their condition; these scores and the total LYMPROM© score exhibit moderate correlations with more generic EQ5D-5L measures. Further research will explore relationships between patient-level characteristics and LYMPROM© responses, and extend initial work on its cross-cultural validity.

## Introduction

Patient Reported Outcome Measures (PROMs) help patients communicate their health-related quality of life (QoL) and symptoms [1]. PROMs support clinicians to understand the patient's view and deliver person-centred care, along with opportunities to guide shared decision making [2]. Together, these are key tenets of the Value Based Healthcare (VBH) agenda, which seeks to optimise outcomes for patients within a financially viable model [3].

Lymphoedema is a lifelong condition resulting in chronic oedema (swelling) due to stasis of lymph fluid. Commonly, lymphoedema presents within physical extremities, but can also affect the midline regions [4]. The impact of lymphoedema can be far reaching. In addition to physical impact (such as risk of infection, skin changes, heaviness and pain or discomfort) mental and social wellbeing, including altered body image, depression and social isolation, can also be affected [5–8]. Lymphoedema services support patients across their life course, advocating and supporting the cornerstones of care [9] [Fig 1] to reduce the risk of complications

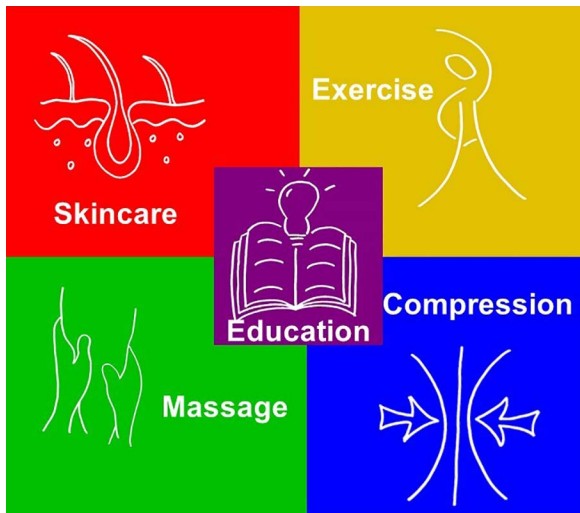

**Fig 1. Cornerstones of lymphoedema care.**

from lymphoedema. Risk reduction, early intervention and ongoing management are costly and labour intensive [10–12]; relying on an effective patient-therapist partnership to understand what is important to the individual to support motivation and engagement [13] in self-management. PROMs are a key part of this, supporting holistic patient assessments and targeted support or specialist interventions to enable prudent use of resources.

Lymphoedema services in Wales currently support almost 25,000 people with lymphoedema, giving a prevalence of 7.3 per 1,000 people, with a mixed aetiology and presentation of upper, lower and midline lymphoedema [14]. This equates to over 44,000 patient activities annually, including 7,835 new patient referrals, representing an incidence of 2.5 per 1,000 people. We developed LYMPROM© to support VBH as part of providing a sustainable service for therapists and patients to focus on what matters most to patients.

## Provenance of LYMPROM©

Since 2015, lymphoedema services in Wales, under the auspices of the Lymphoedema Wales Clinical Network (LWCN), have been collecting outcome measures during routine patient care [Fig 2] using standardised documentation to plan and prioritise direct patient care.

These outcomes, considered important to all patients with lymphoedema, were identified *via* a literature review and with key stakeholders (patients and therapists) examining the potential benefits of lymphaticovenous anastomosis (LVA) during workshop events (2014-2015). In a series of workshop-style events, patient representatives, therapists and academics together and separately explored the lived and reported whole life impact of lymphoedema. Groups also highlighted the need to develop a succinct and easy to use tool. Using these findings, researchers summarised the whole-life impact through 13 high-level items across three

**Impact of Lymphoedema PROMS / Distress Thermometer**

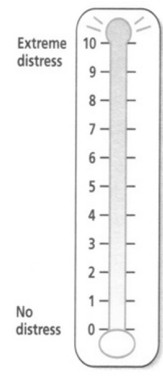

Using the thermometer, write down the number (0 to 10) that best describes how much distress* you have felt in the past week, including today:

*\* Distress is a term used to describe unpleasant feelings or emotions that may interfere with your ability to cope with lymphoedema, its physical symptoms and its treatment. Distress covers a wide range of feelings including anger, frustration, sadness, fear, depression, guilt and anxiety.*

**If the score is more than 5 per question, please consider a referral for counselling**

a.   What is the impact of lymphoedema on your life?  _____

b.   How anxious does your lymphoedema make you feel?  _____

c.   How much does your lymphoedema effect your daily activities (e.g. house work, driving, dressing, cooking)?  _____

d.   What effect does lymphoedema have on your work?  _____

e.   What effect does lymphoedema have on your hobbies?  _____

f.   What effect does lymphoedema have on you when shopping for clothes or shoes?  _____

g.   Does your lymphoedema affect your body image?  _____

h.   Does your lymphoedema affect your sexuality/intimacy?  _____

LNW Adult Lymphoedema Review Sheet            V3.0 14/11/2019            Page **2** of 6

**Fig 2. Example of historic outcome measures collected at each patient contact.**

domains, encompassing Physical health, Social health and Emotional health. For example, washing, dressing, and eating were included within the "personal care" item, a component of Social health. This condensed PROM was then reviewed, and a free text option included to enable patients to describe any other impact of lymphoedema. In 2020, these items were reviewed and extracted into a PROM called LYMPROM© [15]. The free text box enables screening for new themes for further consideration, and work exploring such themes will be reported elsewhere.

Acknowledging current considerations for existing PROMs [16] and a need for a site-agnostic lymphoedema PROM [17,18]; LYMPROM© was developed for use by patients with upper/lower limb or midline lymphoedema.

## The LYMPROM©

LYMPROM© [Fig 3] has 13 closed response items with a free text box for patients to provide additional information. Based on stakeholder's feedback, four items offer a 'not applicable' (N/A) option. All items are equally weighted and scored using an 11-point scale, where zero indicates no impact and 10 indicates an extreme impact. The LYMPROM© score is then reported as a percentage of the maximum possible score, based on the number of items answered (excluding N/A or omitted responses). Higher LYMPROM© scores are thus associated with poorer health.

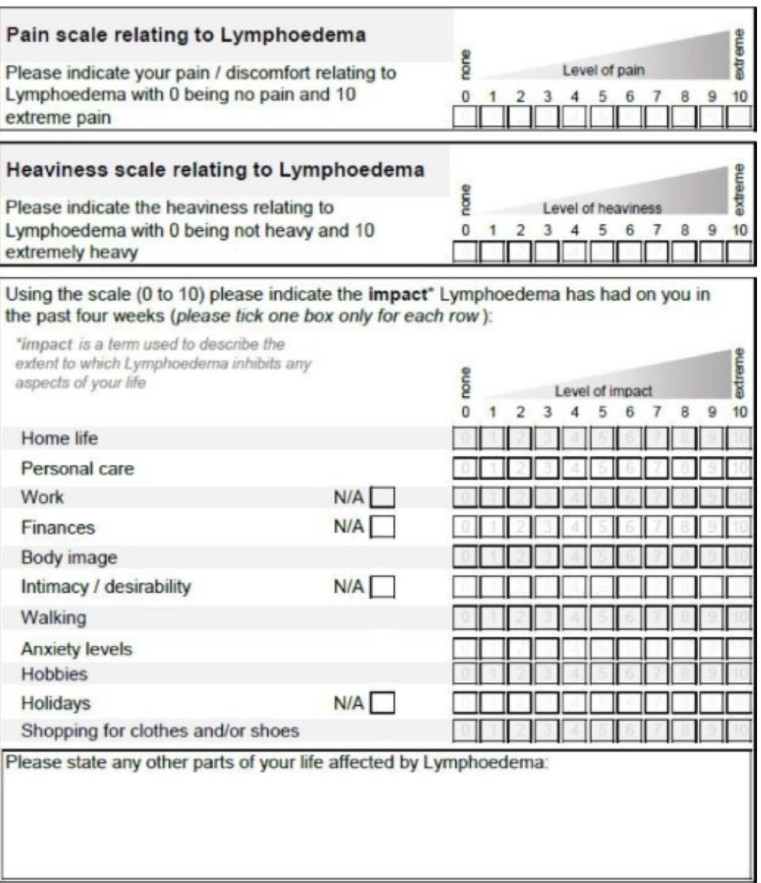

**Fig 3. LYMPROM© and translations are protected by copyright with all rights reserved to Lymphoedema Wales.**

LYMPROM© is available on paper and digitally. In the adaption of the digitised LYM-PROM©, guidelines were followed with cognitive interviews outstanding [19]. To support completion of the paper-based LYMPROM© as a one-page document, guidance information was placed overleaf [Figs 3 and 4]. Within the digitised form, guidance text appears with each of the original items [Fig 5]. Across Wales, automated digital platforms are supporting a standardised way of working and communication between patients and therapists [20], whilst enabling timely access to LYMPROM© when a patient is referred to the service and ahead of each planned contact.

## LYMPROM© domains

A reflective model is assumed whereby the items together explore the underlying construct of lymphoedema [Table 1]; the three domains (Physical health; Social health; and Emotional health) each include at least two items with no explicit N/A option. LYMPROM© domain scores are calculated as percentages of the maximum possible score, accounting for both differing numbers of items and any N/A or omitted responses.

## Methods

This paper builds on work previously published [15] and follows the principles set out within the COSMIN (COnsensus-based Standards for the selection of health Measurement

When you complete LYMPROM© focus on your Lymphoedema and its impact on:-
• **Pain** - indicate from 0 to 10 your pain / discomfort / ache due to Lymphoedema
• **Heaviness** - indicate how heavy your limb(s) feels due to Lymphoedema
• **Home life** - includes housework / cooking / social life / family life / driving
• **Personal care** - includes washing / dressing / eating
• **Work** - includes paid & voluntary work as well as activities such as caring for grandchildren or others. Please tick N/A if you are not currently working
• **Finances** - have your finances been affected; are you financially worse off or have you found it difficult to manage financially because of Lymphoedema? Please tick N/A if you are not currently working
• **Body image** - how do you feel about the way Lymphoedema makes you look? You can think of body image as your views (or feelings) on your body
• **Intimacy / desirability** - how attractive do you feel and the effect Lymphoedema has on your intimate relationships. Please tick N/A if you do not want to answer this question
• **Walking** - includes balance, distance, ease of walking, falls
• **Anxiety levels** - does Lymphoedema make you feel anxious? You can think of anxiety as a feeling that includes being worried, tense or afraid. It may also affect you physically
• **Hobbies** - has your Lymphoedema impacted types of hobbies such as sports, knitting, sewing, reading, gardening etc.
• **Holidays** - have you changed holiday patterns / plans / destinations due to your Lymphoedema? This includes short breaks and day trips. Please tick N/A if you do not take holidays
• **Shopping for clothes and / or shoes** - the impact of Lymphoedema on choices and types of clothing/ shoes, willingness to go shopping

**Fig 4. Supporting text on the back page of the paper-based LYMPROM©.**

## Pain scale relating to Lymphoedema

Please indicate your pain / discomfort / ache relating to Lymphoedema with 0 being no pain and 10 extreme pain.

Level of pain - indicate from 0 to 10 your pain/discomfort/ache due to Lymphoedema*

○ 0  ○ 1  ○ 2  ○ 3  ○ 4  ○ 5  ○ 6  ○ 7  ○ 8  ○ 9  ○ 10

**Fig 5. Supporting text with each item on the digitised LYMPROM©.**

**Table 1. Overview of LYMPROM© items and domains.**

| Domain | LYMPROM© item | number | key |
|---|---|---|---|
| Physical health | pain/ discomfort | 1 | PH1 |
| | heaviness | 2 | PH2 |
| | walking | 9 | PH3 |
| Social health | home life | 3 | SH1 |
| | personal care | 4 | SH2 |
| | work* | 5 | SH3 |
| | finances* | 6 | SH4 |
| | hobbies | 11 | SH5 |
| | holidays* | 12 | SH6 |
| | shopping for clothes and/or shoes | 13 | SH7 |
| Emotional health | body image | 7 | EH1 |
| | intimacy/ desirability* | 8 | EH2 |
| | anxiety | 10 | EH3 |

* denotes an item with an explicit N/A response option.

INstruments) guidelines [21,22]. Four linked phases of the study were undertaken with data collected from late 2020 to early 2022. Different participants aged 18 years or older with lymphoedema were involved in each phase of testing. All participants continued to receive usual care with LYMPROM© available in English and Welsh. All data were anonymised.

## Ethical considerations

This study was reviewed at the Research and Development Joint Scientific Review Committee at Swansea Bay University Health Board as a service evaluation (14/04/2020). All data used in this study was anonymous and involved adults (aged 18 or older) participants only. Phase one of this study involved an anonymous survey using staff mailing lists and voluntary stakeholder group contacts. All participants were made aware of the voluntary nature of the study, its purpose and use of data. As part of service evaluation, consent was not required, with no risk of harm identified for participants and options for participants to opt out freely.

The LWCN privacy notice was made available to all participants of this study, outlining the use of anonymised data in routine evaluations and research. To support the process of data collection for phase three (test-retest) all participants were sent an information sheet ahead of their usual appointment explaining the study purpose, data use and voluntary nature of completing an additional LYMPROM© before their usual appointment (i.e., without any impact on clinical care). As outlined to potential participants, consent was assumed by return of the survey/ additional LYMPROM©. In other phases all data were routinely collected and then anonymised.

LWCN funded this validation study; there was no external funding

## Phase one

Phase one explored content validity from the perspective of patient representatives and healthcare staff; and structural validity and internal consistency using anonymised routinely collected data. Translation of LYMPROM© from English into Welsh followed guidelines [23] to promote rigour and ensure the most appropriate translated phrases/ words were used to convey the original intention of the question. Welsh Language legislation makes this

translation a legal requirement for public bodies in Wales [24]. Translational validity is briefly reported in this paper.

### a: Content validity

LWCN staff and patient support groups, as experts in lymphoedema, were invited, via a short explanatory email, to share their views on LYMPROM©. Patient support groups were accessed to gather patient representative views outside of NHS care providers. Anonymous feedback, returned by post/ email, covered the relevancy, clarity and comprehensiveness of LYM-PROM© items [15], using a four-point ordinal scale of agreement (from "strongly disagree" to "strongly agree") to rate each item for both relevancy and clarity. Respondents were also asked to assess LYMPROM© in overall terms, and could use a free text box to explain responses; for example, to describe items that were scored as low. Alongside the item scores, this information was used to inform item revision, removal or addition.

### b: Translational validity

LYMPROM© was originally developed in English and translated into Welsh. Two independent forward translations were undertaken with mother tongue in the forward translation language (Welsh). One of the forward translators was an expert in lymphoedema, the construct under investigation. One back translation was used with a person with a mother tongue in the backward translation language (English), but non-expert in lymphoedema. All translators worked independently. Translations were collated by a researcher (MGW) who identified discrepancies and provided a feedback report. All discrepancies were discussed with the translators to resolve any issues.

Low use of LYMPROM© Cymraeg precluded formal assessment of its cross-cultural validity, which seeks to examine if a measure behaves similarly in different populations - for example in different ethnic groups or cultures. Examination of cross-cultural validity, and further translation into other languages, are included in objectives for further research.

### c: Structural validity and internal consistency

Anonymised LYMPROM© data were collected over seven months in three Welsh Health Boards. From this data, we reported item floor or ceiling effects (the percentage of respondents that record the lowest or highest possible score), item correlations, summaries of LYM-PROM© domain and total scores, including correlations.

### Phase two: construct validity

As part of routine care, participants completed EQ5D-5L and LYMPROM© questionnaires on the same date. Convergent and divergent validity were assessed by examining correlations between LYMPROM© total and domain scores and the EQ5D-5L utility scores and thermometer readings.

### Phase three: test-retest reliability

A test-retest study alongside usual care was undertaken in one Health Board over two months, with recruitment reviewed at six weeks. An information sheet was posted to patients in the month before their scheduled appointment explaining the voluntary nature of this study, which would report only anonymised results. Patients were asked to complete an additional LYMPROM© during the two weeks before their usual appointment, and explicitly reminded that they could freely choose not to complete the 'additional' LYMPROM© for the study without impact on their usual care.

The interval of two weeks between LYMPROM© administrations was chosen to reduce the risk of recall or changes in lymphoedema symptoms. Reliability was assessed through change in and correlations between the two sets of LYMPROM© domain and total scores.

### Phase four: responsiveness

Longitudinal validity examined changes in LYMPROM© scores after patients received their first intervention for lymphoedema, using the LYMPROM© completed as part of triage (when patients are referred to the service) and the first LYMPROM© after follow-up, between four weeks and seven months later. This follow-up interval provided enough time for outcomes to change, with an expected improvement in outcomes (hence *lower* LYMPROM© scores) based on therapists supporting participants with the cornerstones of care [Fig. 1.]. Those followed-up within four weeks were excluded, as there would been insufficient time for any intervention effect to be observed; the limit of seven months reduced the risk of patients initially improving but subsequently deteriorating. We were unable to explore compliance with care.

### Sample size considerations

For an adequate assessment of content validity [Phase 1a], the target sample size was 30-49 participants, based on guidelines for content validity within survey studies [22]. For each of Phases 1c, 2-4, the target sample size, reflecting guidelines for assessing construct validity, test-retest reliability, and responsiveness, was 100 participants [22], which represents more than seven participants per LYMPROM© item.

### Data management and statistical methods

Participant demographic data and questionnaire responses were initially stored in EXCEL spreadsheets; anonymised data were subsequently imported into SPSS (version 29) for further processing, summary and analysis. We did not plan to impute any missing data.

We calculated a Content Validity Index (CVI, or item-CVI) as the proportion of experts agreeing or strongly agreeing that each of the 13 LYMPROM© items were relevant and clear, excluding any missing responses. We also calculated a Content Validity Index (CVI) to report the proportion of respondents that "agreed" or "strongly agreed" with a range of overall aspects of LYMPROM©, and reported the average of these CVIs. We used the following threshold guidelines values: 0.78 for accepting an item CVI; and 0.9 for the average CVI [25–27].

We used standard Pearson correlation coefficients to summarise relationships between PROM scores, with interpretation based on the following (absolute) thresholds: strong correlation > 0.7, moderate 0.5-0.7, weak 0.3-0.5, very weak < 0.3 [28], further informed by anticipated strengths of correlation. We used linear regression models to assess changes in LYMPROM© total and domain scores as functions of time (in days) between initial and repeat scores. We summarised changes using a change per day regression coefficient, and the correlation between change and time.

## Results

### Recruitment & Participant Characteristics

Across the four phases of the study, 732 participants were recruited [Table 2]. This total included 32 (mainly female) respondents contacted *via* LWCN mailing lists to complete the CVI [Table 3,4].

For Phase 1a, missing items were sparse and there were no commonly missed responses. For Phases 1c & 2, responses for items with no explicit N/A were largely complete (>98%);

**Table 2. Characteristics of participants.**

| Phase | 1a<br>Content validity | | 1c & 2<br>Structural & Construct validity | | 3<br>Test retest reliability | | 4: Responsiveness | |
|---|---|---|---|---|---|---|---|---|
| Participants | [n=32] | | [n=433] | | [n=132] | | [n=135] | |
| **Characteristic** | | | | | | | | |
| **Gender** | n | (%) | n | (%) | n | (%) | n | (%) |
| Female | 28 | (87.5%) | 245 | (56.6%) | 80 | (60.6%) | 84 | (62.2%) |
| Male | 3 | (9.4%) | 161 | (37.2%) | 51 | (38.6%) | 37 | (27.4%) |
| Other/ prefer not to say | 0 | (0%) | 0 | (0%) | 1 | (0.8%) | 1 | (0.7%) |
| Not reported | 1 | (3.1%) | 27 | (6.2%) | 0 | (0%) | 13 | (9.6%) |
| **Age (years)** | n | (%) | n | (%) | n | (%) | n | (%) |
| 18-24 | 1 | (3.1%) | 2 | (0.5%) | 1 | (0.8%) | 0 | (0%) |
| 25-34 | 5 | (15.6%) | 3 | (0.7%) | 2 | (1.5%) | 6 | (4.4%) |
| 35-44 | 6 | (18.7%) | 21 | (4.8%) | 14 | (10.6%) | 16 | (11.9%) |
| 45-54 | 5 | (15.6%) | 30 | (6.9%) | 25 | (18.9%) | 32 | (23.7%) |
| 55-64 | 4 | (12.5%) | 72 | (16.6%) | 36 | (27.3%) | 32 | (23.7%) |
| 65-74 | 2 | (6.2%) | 71 | (16.4%) | 36 | (27.3%) | 39 | (28.9%) |
| 75-84 | 2 | (6.2%) | 107 | (24.7%) | 14 | (10.6%) | 9 | (6.7%) |
| 85 + | 1 | (3.1%) | 107 | (24.7%) | 4 | (3.0%) | 1 | (0.7%) |
| Not reported | 6 | (18.7%) | 20 | (4.6%) | 0 | (0%) | 0 | (0%) |
| **Lymphoedema** | | | n | (%) | n | (%) | n | (%) |
| Upper limb | Not relevant | | 14 | (3.2%) | 16 | (12.1%) | 10 | (7.4%) |
| Lower limb | Not relevant | | 388 | (89.6%) | 93 | (70.5%) | 91 | (67.4%) |
| Midline | Not relevant | | 2 | (0.5%) | 9 | (6.8%) | 7 | (5.2%) |
| Multiple areas | Not relevant | | 7 | (1.6%) | 14 | (10.6%) | 14 | (10.4%) |
| Not reported | Not relevant | | 22 | (5.1%) | 0 | (0%) | 13 | (9.6%) |

**Table 3. Content validity: results.**

| LYMPROM© item | n | Content Validity Index | |
|---|---|---|---|
| | | **Relevant** | **Clear** |
| Pain | 32 | 0.97 | 0.94 |
| Heaviness | 32 | 1.00 | 0.97 |
| Home Life | 32 | 1.00 | 1.00 |
| Personal care | 32 | 1.00 | 1.00 |
| Work | 31 | 0.94 | 0.90 |
| Finances | 29 | 0.83 | 0.90 |
| Body image | 30 | 0.97 | 0.97 |
| Intimacy/desirability | 31 | 0.94 | 0.90 |
| Walking | 32 | 1.00 | 0.97 |
| Anxiety | 32 | 0.94 | 0.97 |
| Hobbies | 32 | 0.97 | 0.97 |
| Holidays | 31 | 0.87 | 0.94 |
| Shopping for clothes and/or shoes | 32 | 0.97 | 0.97 |
| Free text box | 32 | 1.00 | 1.00 |

in contrast, responses rates on the four items with an explicit N/A option did not exceed 25%. All LYMPROM© data were returned in full for phases three and four. Characteristics of participants in these phases were broadly comparable, and significantly younger than those in

**Table 4. Content validity: assessment.**

| LYMPROM© overall review | n | Content Validity Index |
|---|---|---|
| Covers all outcomes that are important to patients | 30 | 0.97 |
| Covers all outcomes that are important to clinicians | 31 | 0.87 |
| The order of the questions is appropriate | 31 | 0.94 |
| The response options are clear | 31 | 0.94 |
| The response options are relevant | 31 | 1.00 |
| It is easy to complete | 30 | 0.97 |
| It looks appealing | 32 | 0.88 |
| There are overlapping questions | 31 | 0.48 |
| The information for use is relevant | 30 | 1.00 |
| The information for use is clear | 30 | 0.94 |
| The information for use appears in the correct place | 30 | 0.87 |
| The information for use is the right length | 31 | 0.97 |
| **Average Content Validity Index** | | **0.94** |

phase two. Most participants identified as female, and more than two-thirds reported lower limb lymphoedema.

## Content validity [Phase 1a]

The content validity data presented here builds on previous studies [15]. Item CVIs for relevance and clarity were all in the range from 0.83 to 1.00; the average CVI, an overall assessment of aspects of LYMPROM©, was 0.94, with one aspect below the threshold (presence of overlapping items, with CVI = 0.48). [Table 3,4]. Respondents were asked to provide further detail for any items they rated lowly; in total 20/32 (63%) provided free text comments to explain their scoring and review of LYMPROM© covering further details on:
Relevancy

- Ensuring all relevant information was included in item descriptions (reports of discomfort or ache suggested were added to pain)

- Some items were considered not applicable to all patients (finances and holidays, with a not-applicable option added)

- Potential overlap of items

Comprehensiveness

•No items were found to cover outcomes related to compression garments or wounds

Appearance

- The form was considered by some as lengthy (too long, asking for too much, busy appearance)

- Presentation and in particular having the instructions for use on back page for paper LYMPROM© forms

## Translation [Phase 1b]

The LYMPROM© translations from English to Welsh and back (two independent forward and one back translations) identified minimal discrepancies. The term used to denote midline ("*corff*") was reviewed and agreed before being separated to denote specific midline regions of the body.

The forward-back translation of "intimacy" to "*agosatrwydd*" returned the term "proximity". Upon review, all agreed that the translations maintained the meaning of original word: intimacy. Similarly, a single Welsh term was used for "pain" and "ache" as the terms were synonymous.

## Structural validity [Phase 1c]

In total, 433 sets of responses were included in the analysis, with items marked as not-applicable recorded as missing and excluded from analyses. Using paper-based questionnaires resulted in relatively few missing responses. Summaries for LYMPROM© items (including floor and ceiling effects) [Table 5]; and summaries for the LYMPROM© total and domain scores [Table 6], for all participants and by gender and (conflated) age bands were reported.

## Phase two. construct validity

We first considered correlations between LYMPROM© items; and then reported correlations between LYMPROM© total and domain scores; and, finally, we gave the correlations between LYMPROM© total and domain scores with EQ5D utility scores and the "thermometer" or visual analogue scale (VAS) readings. We noted that LYMPROM©, a condition-specific PROM, was constructed so that higher scores were associated with poorer health, while EQ5D, a general QoL measure, had higher scores associated with better health, the two measures (and sub-domains) were expected to be negatively correlated.

**Table 5. LYMPROM Item scores summaries; floor and ceiling effects.**

| | Responses | Item Score | | Floor & ceiling effects | |
|---|---|---|---|---|---|
| **LYMPROM© Item** | **n (missing)** | **Mean (SD)** | **Median (LQ, UQ)** | **Best health (score = 0)** | **Worst health (score = 10)** |
| | | | | **n(%)** | **n(%)** |
| Pain | 433 (0) | 3.84 (2.96) | 4 (1, 6) | 77 (17.8%) | 18 (4.2%) |
| Heaviness | 432 (1) | 4.88 (2.86) | 5 (2,7) | 46 (10.6%) | 25 (5.8%) |
| Home life | 432 (1) | 4.38 (3.03) | 5 (2, 7) | 67 (15.5%) | 23 (5.3%) |
| Personal care | 431 (1) | 3.16 (2.97) | 2 (0, 5) | 132 (30.6%) | 10 (2.3%) |
| Work | 61 (372) | 3.48 (3.29) | 3 (0, 6) | 18 (29.5%) | 4 (6.6%) |
| Finances | 54 (379) | 1.78 (3.20) | 0 (0, 2.25) | 37 (68.5%) | 2 (3.7%) |
| Body image | 428 (5) | 3.25 (3.38) | 2 (0, 5) | 159 (37.1%) | 31 (7.2%) |
| Intimacy/ desirability | 107 (326) | 3.28 (3.76) | 2 (0, 6) | 46 (43.0%) | 14 (13.1%) |
| Walking | 432 (1) | 5.20 (3.44) | 5 (2, 8) | 79 (16.3%) | 59 (13.7%) |
| Anxiety | 431 (2) | 3.00 (3.19) | 2 (0, 5) | 149 (34.6%) | 21 (4.9%) |
| Hobbies | 429 (4) | 2.68 (3.10) | 2 (0, 5) | 182 (42.4%) | 18 (4.2%) |
| Holidays | 143 (290) | 3.20 (3.40) | 2 (0, 5) | 54 (37.8%) | 10 (7.0%) |
| Shopping for clothes/shoes | 433 (0) | 4.56 (3.78) | 5 (0, 8) | 126 (29.1%) | 71 (16.4%) |

**Table 6. LYMPROM© total and domain scores.**

| All respondents | | N | Mean | (SD) | Median | (LQ, UQ) |
|---|---|---|---|---|---|---|
| Total score | | 433 | 38.3 | (24.5) | 35.6 | (18.3, 55.0) |
| Physical health | | 433 | 46.4 | (25.6) | 46.7 | (26.7, 66.7) |
| Social health | | 433 | 36.2 | (26.2) | 34.0 | (14.6, 55.5) |
| Emotional health | | 433 | 30.8 | (29.9) | 25.0 | (5.0, 50.0) |
| **By gender** | | | | | | |
| Total score | Female | 245 | 40.5 | (24.1) | 38.3 | (21.7, 58.9) |
| | Male | 161 | 32.8 | (23.7) | 30.8 | (14.4, 46.1) |
| Physical health | Female | 245 | 48.7 | (25.4) | 50.0 | (28.3, 70.0) |
| | Male | 161 | 42.3 | (25.1) | 43.3 | (26.7, 58.3) |
| Social health | Female | 245 | 38.2 | (25.3) | 35.0 | (17.8, 57.5) |
| | Male | 161 | 31.3 | (26.2) | 27.1 | (7.5, 45.5) |
| Emotional health | Female | 245 | 33.6 | (29.9) | 25.0 | (5.0, 55.0) |
| | Male | 161 | 22.4 | (27.0) | 10.0 | (0.0, 37.5) |
| **By age band** | | | | | | |
| Total score | 18-54 | 56 | 44.2 | (27.2) | 41.5 | (23.5, 64.3) |
| | 55-64 | 72 | 41.5 | (28.5) | 41.5 | (15.7, 61.9) |
| | 65-74 | 71 | 34.0 | (25.7) | 29.0 | (11.1, 51.1) |
| | 75-84 | 107 | 38.6 | (22.6) | 36.7 | (18.9, 57.8) |
| | 85 + | 107 | 33.3 | (18.5) | 32.2 | (20.0, 42.2) |
| Physical health | 18-54 | 56 | 45.1 | (28.5) | 40.0 | (24.2, 70.0) |
| | 55-64 | 72 | 46.6 | (30.1) | 50.0 | (17.5, 70.0) |
| | 65-74 | 71 | 43.9 | (26.0) | 43.3 | (20.0, 63.3) |
| | 75-84 | 107 | 48.0 | (24.5) | 53.3 | (30.0, 66.7) |
| | 85 + | 107 | 46.4 | (20.3) | 46.7 | (33.3, 60.0) |
| Social health | 18-54 | 56 | 41.0 | (28.4) | 39.3 | (17.5, 62.5) |
| | 55-64 | 72 | 38.4 | (28.7) | 37.1 | (10.5, 61.9) |
| | 65-74 | 71 | 32.1 | (28.2) | 24.0 | (5.0, 57.5) |
| | 75-84 | 107 | 37.2 | (24.9) | 35.0 | (17.5, 57.5) |
| | 85 + | 107 | 31.8 | (21.3) | 30.0 | (16.0, 45.0) |
| Emotional health | 18-54 | 56 | 48.9 | (32.7) | 47.5 | (24.2, 77.5) |
| | 55-64 | 72 | 40.9 | (33.4) | 40.0 | (10.0, 65.0) |
| | 65-74 | 71 | 24.4 | (27.5) | 15.0 | (3.3, 40.0) |
| | 75-84 | 107 | 28.1 | (25.4) | 25.0 | (5.0, 40.0) |
| | 85 + | 107 | 17.1 | (20.9) | 10.0 | (0.0, 25.0) |

## A: correlations between LYMPROM© items

Correlations between LYMPROM© items [S1 Table] used the LYMPROM© item order and included 95% CIs and the number of pairs, while [S2 Table] covered items in order of domains. We summarised these strong item correlations *via* the average and range of item correlations within and across domains [Table 7], and note that all item correlations were positive, and in the range 0.376 to 0.827; with more than two thirds (55/78) above 0.5, and 10 above 0.7.

## B: correlations between LYMPROM© total and domain scores

The set of strong, positive item correlations resulted in a further set of positive correlations between the LYMPROM© total and domain scores [Table 8]; the strong correlations

**Table 7.  LYMPROM© item correlations, within and across domains.**

|  | PH | SH | EH |
|---|---|---|---|
| PH Mean correlation | 0.526 |  |  |
| (Minimum, Maximum) | (0.476, 0.625) |  |  |
| [n] | [n=3] |  |  |
| SH Mean correlation | 0.569 | 0.588 |  |
| (Minimum, Maximum) | (0.376, 0.708) | (0.420, 0.773) |  |
| [n] | [n=21] | [n=21] |  |
| EH Mean correlation | 0.453 | 0.588 | 0.731 |
| (Minimum, Maximum) | (0.399, 0.491) | (0.421, 0.737) | (0.658, 0.827) |
| [n] | [n = 9) | [n=21] | [n=3] |

**Table 8.  LYMPROM© total and domain correlations.**

|  |  | Total | PH | SH | EH |
|---|---|---|---|---|---|
| Total | Correlation (95% CI) [n] | – |  |  |  |
| PH | Correlation (95% CI) [n] | 0.888 (0.867, 0.907) [433] | – |  |  |
| SH | Correlation (95% CI) [n] | 0.968 (0.962, 0.974) [433] | 0.812 (0.777, 0.842) [433] | – |  |
| EH | Correlation (95% CI) [n] | 0.847 (0.818, 0.872) [433] | 0.595 (0.531, 0.653) [433] | 0.768 (0.726, 0.804) [433] | – |

between the total and domains scores were expected, since the former is closely based on the latter. We noted that the lowest correlation (0.595, between Physical health and Emotional health domains) suggested that these domains captured different aspects of overall Quality of Life.

## C: correlations between LYMPROM© and EQ5D-5L scores

The observed data confirmed initial expectations of strength and direction of correlations between LYMPROM© domain and total scores with EQ5D-5L utility scores and thermometer readings: moderate (negative) correlations between the LYMPROM© total score and both EQ5D-5L measures, and moderate to weak (negative) correlations between the three domain scores and EQ5D-5L measures. [Table 9].

## Phase three. Measurement error and test-retest reliability

One hundred and thirty-two participants completed two LYMPROM© questionnaires within the specified timeframe of two weeks. Data collected during this period supported the expectation that participants would generally not experience significant changes in their lymphoedema in this timeframe. We assessed test-retest reliability *via* correlations between the LYMPROM© total and domain scores at the two time points [Table 10]. We observed a strong positive correlation of 0.919 between the two LYMPROM© total scores [Fig 7]. Correlations between the three domain scores for Physical health, Social health and Emotional health were also strong and positive, with observed values of 0.922, 0.889 and 0.820, respectively.

**Table 9. Correlations between LYMPROM© total and domain scores and EQ5D-5L measures.**

| | | EQ5D-5L measure | | | |
|---|---|---|---|---|---|
| | | Utility score | | Thermometer reading | |
| | | Correlation [n=430] | 95% CI | Correlation [n=423] | 95% CI |
| LYMPROM© score | Total | -0.577 | (-0.636, -0.510) | -0.519 | (-0.586, 0.446) |
| | Physical health | -0.562 | (-0.623, -0.493) | -0.479 | (-0.549, -0.402) |
| | Social health | -0.540 | (-0.604, -0.470) | -0.496 | (-0.565, -0.421) |
| | Emotional health | -0.465 | (-0.536, -0.387) | -0.435 | (-0.509, -0.355) |

The scatter plot [Fig 6] illustrates the strength of the correlation between LYMPROM© total scores and EQ-5D-5L utility scores.

**Table 10. LYMPROM© total and domain scores for test-retest participants.**

| | | Mean (SD) | Median (LQ, UQ) | Correlation (95% CI) |
|---|---|---|---|---|
| **All respondents [n = 132]** | | | | |
| Total score | Test | 55.2 (25.7) | 60.0 (33.6, 77.5) | 0.919 (0.887, 0.942) |
| | Retest | 54.9 (26.7) | 57.6 (34.0, 77.3) | |
| | Change | -0.2 (10.6) | 0 (-6.5, 4.6) | |
| Physical health | Test | 58.2 (25.4) | 63.3 (37.5, 79.2) | 0.922 (0.892, 0.944) |
| | Retest | 58.5 (27.2) | 63.3 (40.0, 80.0) | |
| | Change | 0.3 (10.5) | 0 (-3.3, 6.7) | |
| Social health | Test | 52.0 (27.5) | 57.1 (26.9, 75.3) | 0.889 (0.846, 0.920) |
| | Retest | 51.4 (28.5) | 54.0 (28.1, 72.5) | |
| | Change | -0.6 (13.3) | 0 (-9.6, 6.4) | |
| Emotional health | Test | 57.7 (30.6) | 61.7 (30.0, 84.6) | 0.820 (0.755, 0.869 |
| | Retest | 57.5 (29.3) | 60.0 (35.4, 83.3) | |
| | Change | -0.2 (18.0) | 0 (-6.7, 5.0) | |

## Phase Four. Responsiveness

A second (repeat) LYMPROM© was completed by 135 participants between four weeks and seven months of being triaged as part of usual care (when an initial LYMPROM© had been completed). The average time between initial and repeat scores was 85.4 days (with a standard deviation of 40.7 days) within the range of 28 days to 209 days.

Summary scores and change in scores, defined as repeat-minus initial, are provided [Table 11]. For LYMPROM© total scores, there was an average reduction of (approximately) 3 units, equating

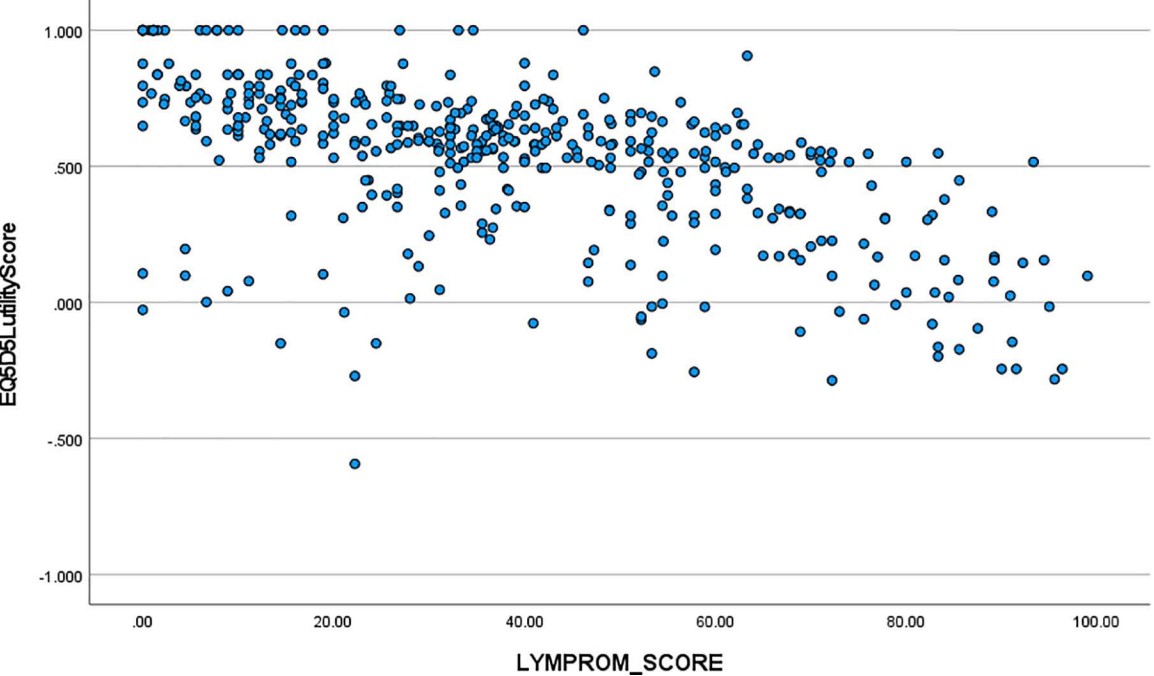

**Fig 6. Scatter Plot of LYMPROM© total and EQ-5D-5L utility scores.**

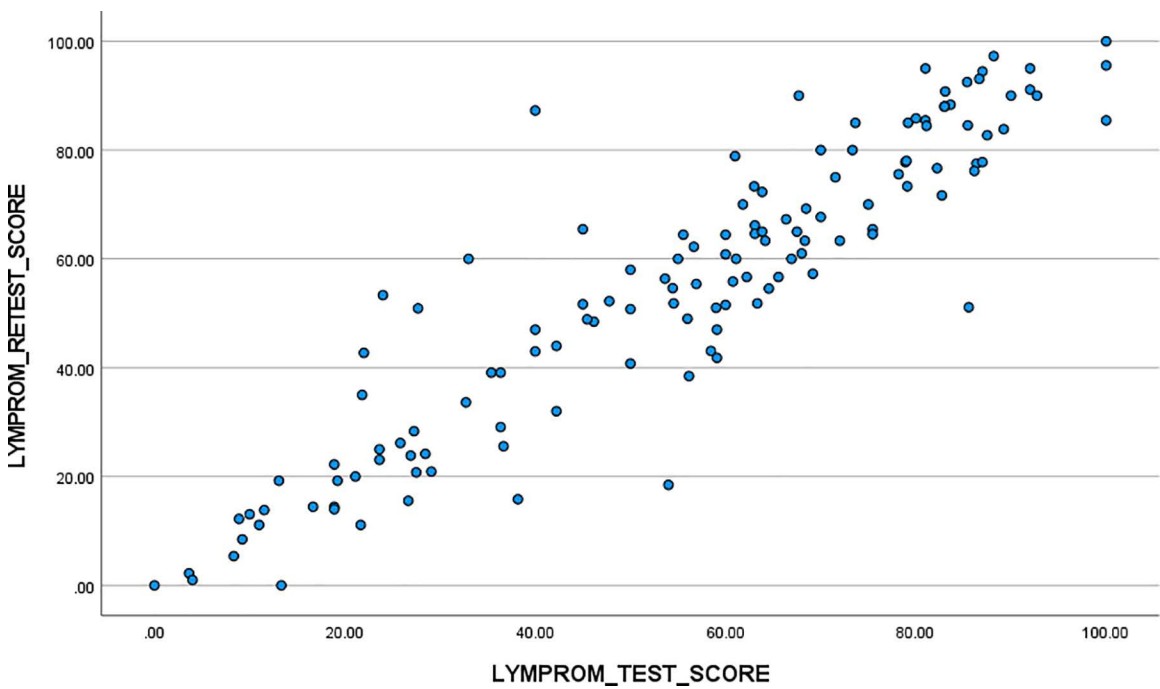

**Fig 7. LYMPROM© test & retest total scores.**

Table 11. LYMPROM© participant total and domain scores for responsiveness.

| LYMPROM component | | Mean | (SD/SE) | Correlation (95% CI) p-value |
|---|---|---|---|---|
| All respondents [n = 135] | | | | |
| Total score | Initial | 57.7 | (SD = 24.2) | -0.193 (-0.350, 0.025) p = 0.025 |
| | Repeat | 54.8 | (SD = 27.5) | |
| | Change | -3.0 | (SD = 18.3) | |
| | Change per day | -0.087 | (SE = 0.038) | |
| Physical health | Initial | 60.7 | (SD = 24.6) | -0.183 (-0.341, -0.014) p = 0.034 |
| | Repeat | 58.1 | (SD = 28.7) | |
| | Change | -2.6 | (SD = 19.4) | |
| | Change per day | -0.087 | (SE = 0.041) | |
| Social health | Initial | 55.3 | (SD = 26.4) | -0.116 (-0.279,0.054) p = 0.181 |
| | Repeat | 51.5 | (SD = 28.3) | |
| | Change | -3.8 | (SD = 20.6) | |
| | Change per day | -0.059 | (SE = 0.044) | |
| Emotional health | Initial | 59.0 | (SD = 30.0) | -0.225 (-0.379, -0.058) p = 0.009 |
| | Repeat | 57.1 | (SD = 31.9) | |
| | Change | -2.0 | (SD = 23.4) | |
| | Change per day | -0.129 | (SE = 0.049) | |

SD = standard deviation; SE = standard error.

to 0.087 units per day, but with large variation; the correlation between change and time of -0.193 (p = 0.025) indicated that this relationship was weak but of borderline statistical significance.

A weak, negative relationship between change in total score and time between scores, with greater (negative) changes generally associated with longer time intervals was confirmed visually [Fig 8].

## Discussion

### Context

PROMs can enhance clinician-patient communication, and whilst many PROM tools exist, relatively few are used when delivering direct patient care [29]. LYMPROM© is one of the first Lymphoedema-specific PROMs (available in English and Welsh) to be validated for adults with lymphoedema, regardless of aetiology or location of lymphoedema. LYMPROM© was created by key stakeholders with vast experience, helping to support feasibility and acceptability in routine clinical care. Using a four phased study different aspects of validity and reliability were explored [22]. In the absence of a gold-standard for comparison, the authors hypothesised weak to moderate negative correlations between EQ5D-5L utility scores and thermometer (or VAS) readings and LYMPROM© Total and Physical health domain scores. Similarly, weak correlations were also expected with scores on the Social health and Emotional health domains.

The characteristics of participants in each study phases were similar [Table 2], however, those involved in phases three and four were younger. Those involved in phase 1a were predominantly female. This is not considered to materially affect results, but will be explored in future work examining LYMPROM© data. Possible explanatory factors include the use in phase one of both paper and digital LYMPROM©, whilst phases three and four used the digital version only, with data collected *via* an automated platform where paper completions were not necessarily entered digitally. There were relatively small numbers of participants in some

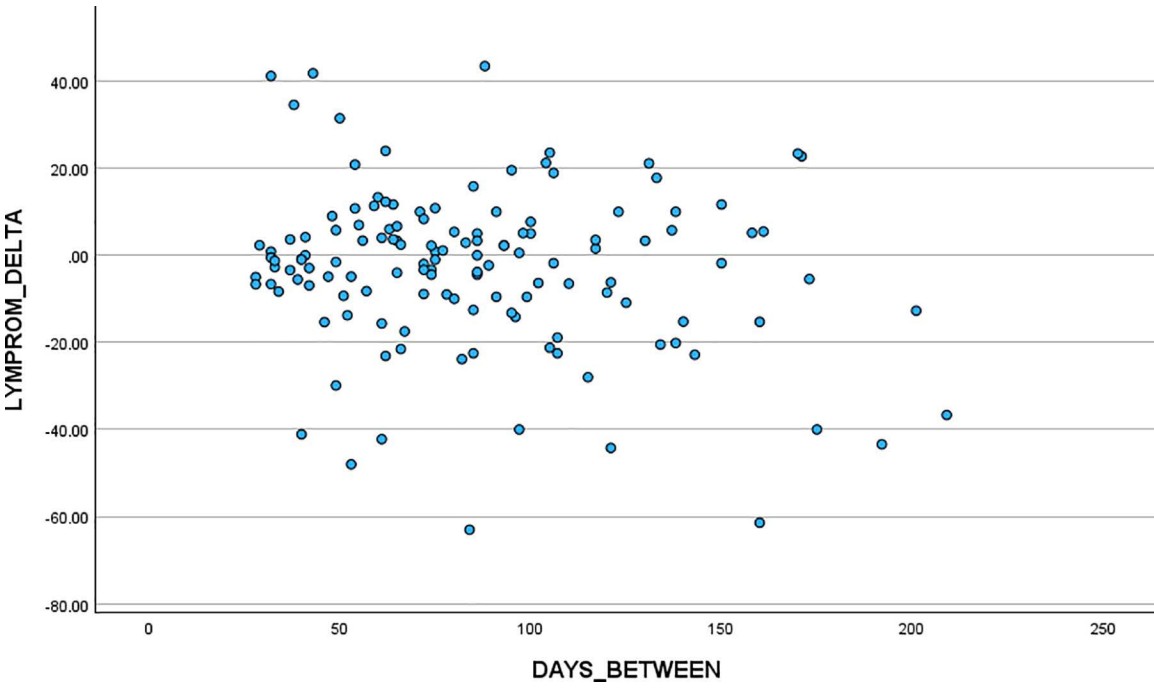

**Fig 8. Scatter Plot LYMPROM© scores. Responsiveness, change of scores over time.**

sub-groups, particularly in phases three and four, which precluded further detailed analyses exploring some potentially interesting patterns observed in the total and domain scores for key factors such as gender or age [Table 6]. Females tended to report higher impact of lymphoedema, with notable peaks in scores for different domains across the age bands; with Emotional health impact being highest for the younger cohorts for example. This interesting point, however, is less relevant to phase 1a, where participants were asked to rate how relevant, clear and comprehensive they felt LYMPROM© was.

## Phase one

The content validity exercise signposted key amendments to LYMPROM©; ensuring pain included reports of discomfort or ache, whilst signalling that not all items would be relevant to all patients (e.g., finances and holidays). Reports of potential overlap of items was in part expected by virtue of the inter-connectedness of some areas such as holidays and hobbies. A small number of survey respondents noted the absence of information on prescribed compression garments or wounds. Compression garment use was considered as an intervention, and wounds as a complication of lymphoedema that would be assessed and reviewed at each patient contact using standardised assessment and review forms. The translation methodology identified important considerations to ensure fidelity with original item wording and intended meaning, which can be further explored in cognitive interviews. Floor effects were observed, with a cohort of patients reporting the best item score (0, zero) [Table 5] Thus LYMPROM© is less able to differentiate those reporting minimal impact of lymphoedema. Positively, this failure to detect variability is not shown for patients reporting worst item scores (10, ten). From a clinical perspective, data reported here includes patients with mild to moderate lymphoedema; rather than patients with severe or complex lymphoedema, who should be the focus of the specialist lymphoedema service. Without clinician-reported outcomes it is difficult to explore this theme further at this time.

## Phase two

Item correlations were generally positive and strong within and across domains [Table 7] and total and domain correlations usually so [Table 8]. This is a key finding; suggesting that the three LYMPROM© domains provide information on discrete aspects of a person. Exploring relationships with EQ5D-5L measures shows a similar correlation with the utility score and thermometer reading [Table 9] and confirms predefined hypotheses. The scatter plot [Fig 6] demonstrates the value of LYMPROM© as a disease specific PROM; capturing discrete aspects of patient outcomes. High utility scores on the EQ5D-5L were reported with a range of LYMPROM© scores, typically below 50. Conversely utility scores of zero on the EQ5D-5L were seen with minimal and extreme impact reported on LYMPROM©. This finding shows that whilst the EQ5D-5L is a good generic measure, the LYMPROM© is a necessary measure to capture lymphoedema specific outcomes.

## Phase three

Test-retest data shows very strong correlations and minimal change over time, with few outliers who do not show consistency in scores [Table 10; Fig 7]. From a clinical point of view, the interval between collections was chosen to reduce any change in the impact of lymphoedema and reflects the assumptions of this phase of the study.

## Phase four

There is a weak negative relationship in scores over time [Fig 8]. The reduction in scores (reduced impact) is seen in the total score and across all domains. The interval period appears to influence the magnitude of score change and this warrants further exploration. For example, larger reductions in scores tended to be seen with longer intervals between LYMPROM© completions within usual care. Moreover, the period of data collection, after the Covid-19 pandemic is an important consideration as this represents a time of recovery to business as usual and the provision of in-person contacts for all. These findings represent a positive clinical outcome, where patients report a reduced impact of their lymphoedema as part of usual care and provides insight into ongoing work to establish what constitutes the minimum clinically important difference and collection of clinician reported outcomes with PROMs.

## The strengths of LYMPROM© and opportunities for further work

This study has demonstrated rigour in the development of the LYMPROM© and shown the strength of correlations and structure of the LYMPROM©, negating at this time the need to further explore factor analysis. The rest-retest phase of the study was one of the most challenging parts to design. Ensuring a minimum interval to reduce the risk of change but also to minimise the influence of recall. Findings indicate that the LYMPROM© is able to detect stability in scores. The study hypothesised a reduction in patient scores after their first contact with the service and this too was a challenge of using routinely collected data alone. Whilst significant results were found, the correlations were low, with interesting caveats identified based on the interval (days) between PROM scores. More recent papers have, however, shown that discrete lymphoedema interventions can evoke a greater magnitude of improvement (reduction of LYMPROM© item and total scores) [30,31]. Work is continuing to explore the interesting differences in scores for certain patient subgroups, alongside other factors such as clinician reported outcomes and other socioeconomic factors including ethnicity.

LYMPROM© is shown to support quality care that is informed by patient priorities and values, offering an insight into the prudent use of resources as part of VBH initiative [29].

PROMs and, more widely, Patient Reported Experience Measures (PREMs) [32] prompt patients to reflect on their care and challenge how they view their condition and their role in self-management. Rather than simply retrieving information, PROMs can empower patients to raise important issues with their clinician [1], providing a tool to monitor progress. Key to this work has been the access to an automated digital PROM platform. Being able to complete LYMPROM© at their own time may help patients raise issues they might feel uncomfortable discussing in person. However, more work is needed to support digital access, literacy and uptake, whilst communicating a wider public health message around VBH and patients as key drivers in PROM-led care. Indeed, feedback from patients attending lymphoedema services captured that they were unfamiliar with PROMs in any other part of their healthcare. Ongoing evaluations are driving PROM-led care in Lymphoedema Wales. PROMs are now focussed from the outset within the subjective assessment in the new patient model and reviewed at each contact thereafter. Alongside this, a suite of information resources for patients and therapists is being developed to optimise patient outcomes alongside a programme of emotional and psychological work led by our consultant psychologist.

## Limitations

Data were collected during an extended period reflecting the challenges of collecting data during the Covid-19 pandemic. The study made use of anonymised and routinely collected data without cognitive interviews or measures of (in)stability for the test-retest or the responsiveness phase. Cognitive interviews with stakeholders would be beneficial for more in-depth understanding in the future. As many patients are reporting PROMs as a novel tool in their care, LWCN staff have had to explain the value of PROMs during their first patient contact. Moreover, patients have signposted the challenge of providing scores based on their lymphoedema alone in the presence of comorbidities. Therapists have sometimes reported 'coaching' patients on how to focus their scores on the impact of their lymphoedema, rather than comorbidities, which may mean that subsequent scores may be altered. This could be a consideration in the responsiveness data. Thus, a patient's next PROM score might be improved (or worsened) through an improved understanding of what they should be scoring on the LYMPROM©. Unfortunately, in making use of routinely collected data, there were no reported 'markers' of stability, improvement or worsening by participants during the period of data collection for phases three and four of the study. Moreover, it was not possible to review if participants complied with their plan of care [Fig 1], which could impact on their outcomes.

Findings should also be interpreted with caution owing to the challenges of providing a virtual service to a population with a condition that so closely relies on touch and pressure. Moreover, lymphoedema as a chronic condition is not always quick to react to interventions and once again in lacking clinical reported outcomes, it is difficult to fully explore these findings. Whilst, the authors of this study did not use a measure of stability/ change over time, the effect of change on test-retest appears to be inconsequential.

## Conclusion

This study has gathered evidence to report on the reliability and validity of LYMPROM© as a measurement to help patients communicate the potential impact of lymphoedema. Evidence demonstrates that LYMPROM© has good content and construct validity and is reliable for adults with lymphoedema and the processes undertaken to validate LYMPROM© Cymraeg supported early identification of potential sources of confusion, thus supporting a quality translation.

LYMPROM© affords a unique opportunity for lymphoedema services to use a single PROM for adults as part of VBH to support a patient-centric service that is sustainable and responsive to patient need at that time. This could remove the need to have a bespoke PROM in use for patients with midline or upper/lower limb lymphoedema. Based on feedback, LYMPROM© has empowered patients and enabled therapists, but more work is needed to optimise use and make PROM talk a part of usual care. Providing important patient-centric information as part of the standardised assessment and review forms empowers patients and therapists to care beyond the historical focus on anthropomorphic measures. The next phase of work will explore other important measures including patient experience and clinician reported measures.

## Supporting information

**Table S1. Correlation matrix for LYMPROM© items.**
(DOCX)

**Table S2. Correlation matrix for LYMPROM© items (coded).**
(DOCX)

## Acknowledgments

The authors would like to extend their thanks to the patients, staff and local Health Board teams including Local and National Lymphoedema Services and Value in Health teams, Rafael Cuartielles for contributing to data management, Russell White for setting up the digital form, and to Rhian Noble-Jones, Karen Roberts, and the local translation unit for their support with translating LYMPROM©.

## Author contributions

**Conceptualization:** Marie Gabe-Walters.

**Data curation:** Marie Gabe-Walters.

**Formal analysis:** Melanie Thomas, Marie Gabe-Walters, Alan Watkins.

**Investigation:** Melanie Thomas, Marie Gabe-Walters, Ioan Humphreys, Alan Watkins.

**Methodology:** Melanie Thomas, Marie Gabe-Walters.

**Project administration:** Marie Gabe-Walters.

**Validation:** Marie Gabe-Walters, Ioan Humphreys, Alan Watkins.

**Visualization:** Melanie Thomas.

**Writing – original draft:** Melanie Thomas, Marie Gabe-Walters, Ioan Humphreys, Alan Watkins.

**Writing – review & editing:** Melanie Thomas, Marie Gabe-Walters, Ioan Humphreys, Alan Watkins.

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
