## [Decision Letter · Decision Letter 0]

27 Sep 2024

PONE-D-24-33349A new validated Lymphoedema-specific Patient Reported Outcome Measure (LYMPROM©) for adults with LymphoedemaPLOS ONE

Dear Dr. Thomas,

Thank you for submitting your manuscript to PLOS ONE. After careful consideration, we feel that it has merit but does not fully meet PLOS ONE’s publication criteria as it currently stands. Therefore, we invite you to submit a revised version of the manuscript that addresses the points raised during the review process.

 The reviewers identified this as important research on PROMs. Please address the points of clarification requested by the reviewers below.

We look forward to receiving your revised manuscript.

Kind regards,

Kathleen Bennett

Academic Editor

PLOS ONE

Journal Requirements: When submitting your revision, we need you to address these additional requirements. 1. Please ensure that your manuscript meets PLOS ONE's style requirements, including those for file naming. The PLOS ONE style templates can be found at https://journals.plos.org/plosone/s/file?id=wjVg/PLOSOne_formatting_sample_main_body.pdf and https://journals.plos.org/plosone/s/file?id=ba62/PLOSOne_formatting_sample_title_authors_affiliations.pdf 2. Please note that in order to use the direct billing option the corresponding author must be affiliated with the chosen institute. Please either amend your manuscript to change the affiliation or corresponding author, or email us at plosone@plos.org with a request to remove this option. 3. We note that your Data Availability Statement is currently as follows: All relevant data are within the manuscript and its Supporting Information files. Please confirm at this time whether or not your submission contains all raw data required to replicate the results of your study. Authors must share the “minimal data set” for their submission. PLOS defines the minimal data set to consist of the data required to replicate all study findings reported in the article, as well as related metadata and methods (https://journals.plos.org/plosone/s/data-availability#loc-minimal-data-set-definition). For example, authors should submit the following data: - The values behind the means, standard deviations and other measures reported;- The values used to build graphs;- The points extracted from images for analysis. Authors do not need to submit their entire data set if only a portion of the data was used in the reported study. If your submission does not contain these data, please either upload them as Supporting Information files or deposit them to a stable, public repository and provide us with the relevant URLs, DOIs, or accession numbers. For a list of recommended repositories, please see https://journals.plos.org/plosone/s/recommended-repositories. If there are ethical or legal restrictions on sharing a de-identified data set, please explain them in detail (e.g., data contain potentially sensitive information, data are owned by a third-party organization, etc.) and who has imposed them (e.g., an ethics committee). Please also provide contact information for a data access committee, ethics committee, or other institutional body to which data requests may be sent. If data are owned by a third party, please indicate how others may request data access. 4. We notice that your supplementary tables are included in the manuscript file. Please remove them and upload them with the file type 'Supporting Information'. Please ensure that each Supporting Information file has a legend listed in the manuscript after the references list. 5. Please review your reference list to ensure that it is complete and correct. If you have cited papers that have been retracted, please include the rationale for doing so in the manuscript text, or remove these references and replace them with relevant current references. Any changes to the reference list should be mentioned in the rebuttal letter that accompanies your revised manuscript. If you need to cite a retracted article, indicate the article’s retracted status in the References list and also include a citation and full reference for the retraction notice.

Reviewers' comments:

Reviewer's Responses to Questions

**Comments to the Author**

1. Is the manuscript technically sound, and do the data support the conclusions?

Reviewer #1: Yes

Reviewer #2: Yes

2. Has the statistical analysis been performed appropriately and rigorously? 

Reviewer #1: Yes

Reviewer #2: Yes

3. Have the authors made all data underlying the findings in their manuscript fully available?

Reviewer #1: Yes

Reviewer #2: Yes

4. Is the manuscript presented in an intelligible fashion and written in standard English?

Reviewer #1: Yes

Reviewer #2: Yes

5. Review Comments to the Author

Reviewer #1: This is an interesting and well written manuscript. It is good to see a PROM being used as part of normal clinical care and this manuscript describes validation of the LYMPROM within that context. I have made some comments below that I believe would improve the manuscript:

1.Ethics statement. The R&D committee is the Joint Scientific Review Committee' and not the 'Joint Study Ratification Committee'.

2.Page 4 onwards. LWCN needs to be defined on first use.

3. Introduction. On page 5, the authors have alluded to earlier development of the LYMPROM and although they have provided a reference, I feel this statement is rather vague. It would be useful to summarise how the LYMPROM was initially developed and the decisions made about what was included.

4. LYMPROM domains, page 6. This sentence is not clear: "For construct validity, LYMPROM© domain scores are calculated as percentages of the maximum possible score, which accounts for both differing numbers of items and any N/A or omitted responses". Please explain how this relates to construct validity.

5. Methods, page 7. The dates of the study given in the methods are different from those in the ethics section. Differing dates are given for different phases which is rather confusing. Perhaps consider length of phases with a single study duration at the start of methods section?

6. The manuscript uses 'data' in both the singular and plural (data is/data are). My preference would be plural but singular is acceptable, but the authors need to be consistent throughout the manuscript.

7. Phase one, page 8. The sentence 'translation into Welsh was briefly considered' is rather vague. Elaborate.

8. Content validity, page 8. What do the authors mean by 'Lymphoedema staff mailing lists and representative patient support groups 'aligned' with LWCN were approached?

9. Content validity, page 8. The desired sample size is given along with a reference. It would be useful to have sentence or two explaining the basis for the sample size rather than having to refer to the reference.

10. Content validity, page 8/9. There needs to be a more detailed description of the Content Validity Index.

11. Translational validity, page 9. It's not clear what this sentence means, please elaborate: 'Welsh translation in itself was not considered sufficient to warrant examination of cross-cultural validity'.

12. Phase 3, page 10. The sample size calculation again needs some discussion.

13. Phase four Responsiveness, page 10. This sentence is not clear: 'This interval provided enough time for

outcomes to change without deterioration'. Surely the researchers are looking for change including deterioration?

14. Results, page 11 onwards should be presented as past tense.

15. Figure 1 doesn't really add anything and could be discussed in the text.

Reviewer #2: Thank you for the opportunity to review this work. It was generally very clearly explained and think it is a really useful piece of work and will be a very valuable tool within lymphoedema My only comments are

Phase 4

– line 20 – can you explain the phrase “change without deterioration” as would there not be the possibility of improvement as they have started treatment but also of deterioration if not able to follow the plan or a change in condition?

Data management and statistical methods

- Is there any similarities in what was commonly missing and would this have an impact on results?

- Number of males completing phase 1a is very low – does this impact on content?

- How was the sample number for each phase calculated?

- The explanation in the discussion is very clear and although this is a very useful tool there is still work to be done as many of the correlations carried out were low as explained. Following this should the conclusion in the abstract be worded slightly less strongly to indicate there are limitations made clear by the findings?

6. PLOS authors have the option to publish the peer review history of their article (what does this mean? ). If published, this will include your full peer review and any attached files.

**Do you want your identity to be public for this peer review?** For information about this choice, including consent withdrawal, please see our Privacy Policy .

Reviewer #1: No

Reviewer #2: No

---

## [Author Response · Author response to Decision Letter 1]

21 Nov 2024

Dear Ms Bennett,

Many thanks for the opportunity to responds to the points raised by the academic editor and reviewers. We have found the comments to be very useful and have helped strengthen our manuscript. We have formulated a concise response to each in the table below and have uploaded a tracked and updated version of the manuscript to PLoS One.

Best wishes,

Melanie

Journal requirements Author response

1. Please ensure that your manuscript meets PLOS ONE's style requirements, including those for file naming. We have endeavoured to ensure that the revised manuscript meets PLOS style requirements

2. Please note that in order to use the direct billing option the corresponding author must be affiliated with the chosen institute. Please either amend your manuscript to change the affiliation or corresponding author, or email us at plosone@plos.org with a request to remove this option. The corresponding author has been updated to match the affiliated institute for direct billing.

3. We note that your Data Availability Statement is currently as follows: All relevant data are within the manuscript and its Supporting Information files. We have now made study data available via the Swansea University Open research data Community, hosted by Zenodo.

4. We notice that your supplementary tables are included in the manuscript file. Please remove them and upload them with the file type 'Supporting Information'. Please ensure that each Supporting Information file has a legend listed in the manuscript after the references list. Supplementary tables have been removed from the main manuscript and are now included as Supporting Information.

5. Please review your reference list to ensure that it is complete and correct. If you have cited papers that have been retracted, please include the rationale for doing so in the manuscript text, or remove these references and replace them with relevant current references. Any changes to the reference list should be mentioned in the rebuttal letter that accompanies your revised manuscript. If you need to cite a retracted article, indicate the article’s retracted status in the References list and also include a citation and full reference for the retraction notice. References have been checked, with two additions to cover translation guidelines.

Reviewer 1 comments

Reviewer #1: This is an interesting and well written manuscript. It is good to see a PROM being used as part of normal clinical care and this manuscript describes validation of the LYMPROM within that context. I have made some comments below that I believe would improve the manuscript: Thank you for your valuable comments; we have addressed these as documented below. Page and line numbers refer to our revised manuscript (clean version).

1.Ethics statement. The R&D committee is the Joint Scientific Review Committee' and not the 'Joint Study Ratification Committee'. Many thanks for highlighting this; we have made the necessary correction. [Page 8, lines 10-11]

2.Page 4 onwards. LWCN needs to be defined on first use. This acronym is now defined. Page 5, line 6].

3. Introduction. On page 5, the authors have alluded to earlier development of the LYMPROM and although they have provided a reference, I feel this statement is rather vague. It would be useful to summarise how the LYMPROM was initially developed and the decisions made about what was included. Further text has now been added to summarise the key steps in creating LYMPROM. [Page 5, lines 14-22]

4. LYMPROM domains, page 6. This sentence is not clear: "For construct validity, LYMPROM© domain scores are calculated as percentages of the maximum possible score, which accounts for both differing numbers of items and any N/A or omitted responses". Please explain how this relates to construct validity. We apologise for the lack of clarity, and have amended the text accordingly. [Page 7, line 9]

5. Methods, page 7. The dates of the study given in the methods are different from those in the ethics section. Differing dates are given for different phases which is rather confusing. Perhaps consider length of phases with a single study duration at the start of methods section? We agree that there is the potential for confusion; and have now simplified timings by reference to a single study duration (late 2020 to early 2022) in the first part of the methods only. [Page 8, line 5] Some previous text has now been removed.

6. The manuscript uses 'data' in both the singular and plural (data is/data are). My preference would be plural but singular is acceptable, but the authors need to be consistent throughout the manuscript. Thank you, we have adjusted throughout.

7. Phase one, page 8. The sentence 'translation into Welsh was briefly considered' is rather vague. Elaborate. We have added details to cover the legal requirement for translation; further details for the translation work are then subsequently provided. [Page 9, line 8; Page 9, line 22 – Page 10, line 4]

8. Content validity, page 8. What do the authors mean by 'Lymphoedema staff mailing lists and representative patient support groups 'aligned' with LWCN were approached? We have added further details explaining these contacts groups. [Page 9, lines 12-14]

9. Content validity, page 8. The desired sample size is given along with a reference. It would be useful to have sentence or two explaining the basis for the sample size rather than having to refer to the reference. All sample size considerations have been consolidated into a new short sub-section, with, we hope, the required level of detail. [Page 11, line 22 – Page 12, line 2] Some previous text has now been removed.

10. Content validity, page 8/9. There needs to be a more detailed description of the Content Validity Index. Further details have been added here on data collection, and in the data management and statistical methods section on data processing. [Page 9, lines 11-20; Page 12, lines 8-14]

11. Translational validity, page 9. It's not clear what this sentence means, please elaborate: 'Welsh translation in itself was not considered sufficient to warrant examination of cross-cultural validity'. We now provided further details on why full cross-cultural validity was not possible at this time. [Page 10, lines 5-9]

12. Phase 3, page 10. The sample size calculation again needs some discussion. This is covered by our response to point 9.

13. Phase four Responsiveness, page 10. This sentence is not clear: 'This interval provided enough time for outcomes to change without deterioration'. Surely the researchers are looking for change including deterioration? We have amended the description and discussion, and hope this has now been more clearly described. [Page 11, lines 14-19]

14. Results, page 11 onwards should be presented as past tense. Thank you, we have adjusted the text accordingly.

15. Figure 1 doesn't really add anything and could be discussed in the text. We would prefer to retain this, as it is now also cited in the limitations for phase four. This is the lymphoedema cornerstone of care that describe key parts of treatment.

Reviewer 2

Reviewer #2: Thank you for the opportunity to review this work. It was generally very clearly explained and think it is a really useful piece of work and will be a very valuable tool within lymphoedema My only comments are Thank you for your valuable comments; ; we have addressed these as documented below. Page and line numbers refer to our revised manuscript (clean version).

Phase 4 – line 20 – can you explain the phrase “change without deterioration” as would there not be the possibility of improvement as they have started treatment but also of deterioration if not able to follow the plan or a change in condition? We agree and have now reworded this to provide clarity, both and briefly in the limitations section. [Page 11, lines 14-19; Page 32, lines 5-8]

Data management and statistical methods - Is there any similarities in what was commonly missing and would this have an impact on results? We have provided more details, covering items with and without an explicit N/A option. [Page 13, lines 7-10]

- Number of males completing phase 1a is very low – does this impact on content? This is explored further in the discussion but is not felt to materially affect results. [Page 13, line 18; Page 27, lines 18-21]

- How was the sample number for each phase calculated? We now refer to published guidelines, and included details in a consolidated sub-section on sample sizes. [Page 11, line 22 – Page 12, line2] Some previous text has now been removed.

- The explanation in the discussion is very clear and although this is a very useful tool there is still work to be done as many of the correlations carried out were low as explained. Following this should the conclusion in the abstract be worded slightly less strongly to indicate there are limitations made clear by the findings? Thank you for this comment; we have now extended the conclusion in the abstract, explicitly noting correlations with the generic EQ5D-5L, and outlining areas for further research, including cross-cultural validity.

[Page 3, lines 13-18]

---

## [Editor Report · Decision Letter 1]

25 Nov 2024

A new validated Lymphoedema-specific Patient Reported Outcome Measure (LYMPROM©) for adults with Lymphoedema

PONE-D-24-33349R1

Dear Dr. Humphreys,

We’re pleased to inform you that your manuscript has been judged scientifically suitable for publication and will be formally accepted for publication once it meets all outstanding technical requirements.

Kind regards,

Kathleen Bennett

Academic Editor

PLOS ONE
---

## [Editor Report · Acceptance letter]

PONE-D-24-33349R1

PLOS ONE

Dear Dr. Humphreys,

I'm pleased to inform you that your manuscript has been deemed suitable for publication in PLOS ONE. Congratulations! Your manuscript is now being handed over to our production team.

Kind regards,

on behalf of

Professor Kathleen Bennett

Academic Editor

PLOS ONE